

# A Surrogate Based Optimization Framework to analyse Stall Induced Vibrations

Chandramouli Santhanam[1], Riccardo Riva[2], and Torben Knudsen[1]

[1]Department of Electronic Systems, Aalborg University, Fredrik Bajers Vej 7, 9220 Aalborg Øst, Denmark
[2]Department of Wind Energy, Technical University of Denmark, Frederiksborgvej 399, 4000 Roskilde, Denmark

**Correspondence:** Chandramouli Santhanam (chsa@es.aau.dk)

**Abstract.** Stall Induced Vibrations (SIV) are an important design consideration for wind turbine blade design, especially for large, modern wind turbines which require long and flexible blades. Their severity depends on the inflow conditions, and structural characteristics of the blade, and the study of the parameter space that leads to SIV has a high computational cost due to the aeroelastic simulations involved, and the cost increases drastically with increasing number of variables to be studied. Given the computationally expensive nature of the problem, in this work, a Surrogate-Based Optimization (SBO) framework which uses surrogate models as an alternative to the high fidelity simulations is proposed to explore the behaviour of SIV. The proposed framework is not specific to any type of surrogate model, and uses Delaunay Triangulation to iteratively select samples to refine the surrogate model. The framework is demonstrated to study the occurrence and severity of SIV on the IEA 10MW turbine in a five variable inflow space consisting of wind speed, yaw angle, vertical wind shear, wind veer, and atmospheric temperature. Using the proposed framework, a well-trained surrogate model is developed and used to predict the damping ratio of the first blade edgewise mode in the entire inflow space at a reduced computational cost. Sensitivity analysis of the predicted damping ratio shows that yaw angle is the most influential variable, while temperature is the least influential variable in terms of inflow conditions that can lead to occurrence of SIV. Inflow conditions with a moderate yaw angle (around 10-25 deg), high wind speeds, and moderate to high negative veer are found to lead to severe SIV. This framework is expected to serve as a guiding tool to decide the scope of the more computationally expensive simulations such as high fidelity CFD-based aeroelastic simulations which can provide a more accurate description of SIV.

## 1 Introduction

Stall-Induced Vibrations (SIV) are an aeroelastic instability that might happen when large portions of the wind turbine blade operate in moderate stall (defined in this work as Angle of Attack (AoA) greater than 15 deg but lower than 40 deg), which leads to large internal loads. The typical negative lift – AoA gradient ($\partial C_L / \partial \alpha < 0$) in this region leads to a negative aerodynamic damping in the sections of the blade, which leads to the lift force being in phase with the velocity of the vibrating blade (Dowell, 2015; Zhao, 2016). Although modern pitch-regulated turbines do not operate in stall like the older stall-regulated ones, computations show that SIV can lead to extreme edgewise vibrations in the turbine blades especially during standstill conditions (Skrzypiński and Gaunaa, 2014). Such standstill conditions may exist during installation, and during times when





the turbine is stopped for maintenance, due to extreme wind speeds or breakdown. Apart from the risk that these vibrations can become vigorous, they also contribute to fatigue loading of the blades which leads to a reduction in lifetime of the blades. With the need to lower the Levelised Cost of Energy driving the design towards larger and more flexible wind turbine blades, SIV are an important design consideration as the loads generated might severely damage the blades.

SIV have been studied using the Blade Element Momentum theory (BEM) based solvers in the limited yaw angle range
around moderate stall regions. The Advanced Aerodynamic Tools for Large Rotors (AVATAR) project details a comprehensive SIV study of BEM-based aeroelastic solvers against a higher fidelity CFD based aeroelastic solver (Heinz et al., 2016). The results show that BEM-based solvers tend to over-predict standstill instabilities due to the utilization of static airfoil data. The validity of the BEM-based solvers decreases as we move into the deep stall regions. Though the engineering models suffer such problems in predicting stall behaviour, they offer a huge reduction in the computational cost, while still having respectable
validity in moderate stall regions. But, it can be seen that even with the cost reduction offered by the engineering models, the exploration of the inflow space is still a very computationally expensive process, and the cost increases drastically with the increasing number of dimensions to explore. For such systems with expensive target functions, Surrogate Based Optimization (SBO) is a technique that has been used to explore the input space effectively. In the context of wind turbine design, surrogate models have been used to study SIV (Santhanam et al., 2022), and in other similar applications involving costly evaluations,
such as evaluation of site specific loads (Dimitrov et al., 2018), finding the the optimal blade geometry parameters and control features (Thapa and Missoum, 2022), and study of the effect of multiple parameters on the blade design (Barlas et al., 2021).

In SBO techniques, an approximate but inexpensive alternative to the target function, called the surrogate model is used to establish a quick to evaluate input-output relation between the target function and the input variables. The surrogate models can then be optimized via standard optimization techniques to identify points of interest in the domain. Popular choices of
surrogate models include Polynomial Response Surfaces, Artificial Neural Networks (ANN), Kriging, Radial Basis Functions (RBF) etc. SBO techniques typically employ three key steps: (1) Selection of initial samples for simulation, (2) Construction of a surrogate model and (3) A strategy called 'Adaptive Sampling' to select samples for subsequent simulations. The process is repeated until the problem specific goals are reached. In many SBO algorithms, the adaptive sampling strategy is suited to a specific type of surrogate model, usually stochastic processes such as Kriging. The use of stochastic processes for exploiting an
input space dates back to Kushner (1964), and since then has gained attraction. A popular algorithm that has been successfully applied in many problems is the Efficient Global Optimization (EGO) algorithm proposed by Jones et al. (1998), based on Kriging and a sampling strategy called Expected Improvement suited to Kriging. Other similar adaptive sampling techniques have been developed based on Kriging, such as Probability of Improvement, Mean Squared Error, Confidence Bounds (Han, 2016). Some of these algorithms have been extended to RBF (Bagheri et al., 2017) and ANN (Metta et al., 2021), but as noted
by Garud et al. (2019a), adaptive sampling strategies are predominantly based on Kriging because of the ready availability of the variance estimate along with the prediction. But depending on the nature of the unknown target function, it may be the case that a different choice of surrogate model works better (Bhosekar and Ierapetritou, 2017; Garud et al., 2019b). Furthermore, most of these sampling methods have the primary goal of searching the parameter space for a minimum, while it is also often useful to obtain a well trained surrogate that can be used for purposes other than finding the minima.

The focus of the current work is to propose tools and methods to study SIV in a given inflow space. We propose an SBO framework to study the inflow space that cause SIV, and demonstrate its use to analyse the behaviour of SIV in a five parameter inflow space. To the authors' best knowledge, the number of studies that have focused on the effective exploration of SIV are very few, and as a new contribution, we propose a framework that is independent of the surrogate model type to effectively study the effect of environmental variables on SIV. The article is organised as follows. Section 2 describes the problem statement and scope of this work. Section 3 describes the simulation setup and methods to characterise SIV. Section 4 describes the proposed framework and illustrates the functionality of the framework on standard test functions and the considered SIV problem. Section 5 explains the effect of the considered variables on SIV.

## 2   Problem statement and scope

The aim of this work is to formulate a SBO framework that can, with the optimal number of simulations

1. Help study the effect of inflow and environmental conditions on SIV

2. Identify the conditions that lead to critical SIV in a wind turbine

The proposed SBO framework should not be surrogate-specific, i.e. it should be able to work with any choice of surrogate model. Studies of SIV in wind turbines (Hansen, 2007; Stettner et al., 2016) show that the occurrence and severity of SIV depends on inflow conditions, airfoil characteristics and structural characteristics of the blade. With this motivation, this framework is used to demonstrate the analysis of the influence of the following variables on SIV

– Wind Speed

– Yaw Angle

– Vertical Wind Shear

– Wind Veer

– Temperature

The parameter space, or inflow space, is defined as the five dimensional space made up of these variables.

## 3   Simulation Setup and SIV characterisation

This section describes the turbine model, considerations in the simulation setup and the method used to quantify the effect of SIV. The IEA 10 MW turbine (Bortolotti et al., 2019) is chosen for investigation. The occurrence and characteristics of SIV in this turbine are studied for a parked rotor with a 90 deg pitch angle, 0 deg azimuth angle (blade 1 pointing upwards), constant wind conditions and only the blades are considered flexible. Hence, only the blade modes are considered and not the



turbine ones. The inflow conditions and simulation setup are similar to the considerations in the investigations of SIV in the AVATAR rotor (Heinz et al., 2016). The aeroservoelastic tool `HAWC2` (Larsen et al., 2007) is used to simulate the motion of the wind turbine. The aerodynamics, and the aeroelastic model of `HAWC2` has been verified against full CFD simulations and experiments in previous studies (Madsen et al., 2020) The dynamic stall is modelled through the MHH Beddoes dynamic stall method (Hansen et al., 2004).

The ranges of the considered variables are shown in table 1.

**Table 1:** Ranges of the considered variables

| Variable | Range |
| --- | --- |
| Wind speed | $[25, 60]\ \mathrm{ms^{-1}}$ |
| Yaw Angle | $[0, 40]$ deg |
| $\alpha$ | $[-0.1, 0.4]$ |
| $a_\varphi$ | $[-1.2, 0.5]$ |
| Temperature | $[-15, 20]$°C |

The wind speed range above the cut-out wind speed is considered because it is expected to induce substantial vibrations. The yaw angle range is chosen so as to avoid sections of the airfoil operating in deep stall where the engineering models have limited validity. The shear exponent ($\alpha$) is chosen to cover negative and positive shear. Wind veer is modelled through a coefficient ($a_\varphi$) as proposed in Natarajan et al. (2016).

$$\Delta\varphi(z) = \varphi(z) - \varphi(z_{hub}) \approx a_\varphi e^{-\sqrt{z_{hub}/h_{ME}}} \frac{z - z_{hub}}{\sqrt{z_{hub}h_{ME}}} \left(1 - \frac{z - z_{hub}}{2\sqrt{z_{hub}h_{ME}}} - \frac{z - z_{hub}}{4z_{hub}}\right), \qquad (1)$$

where $z$ represents the height of the point, $\Delta\varphi(z)$ is the veer angle at height $z$, $z_{hub}$ is the hub height, $h_{ME}$ is the modified Ekman atmospheric boundary layer depth, here set to 500 m. In Eq. (1), $a_\varphi$ is a scaling factor to represent the veer, while the last two terms which depend upon the height z give the wind profile. $a_\varphi$ depends on wind parameters and can be considered site-specific, and a recommended range given in Natarajan et al. (2016) is $[-1.2, 0.5]$. As it can be seen from Eq. (1), positive values of $a_\varphi$ correspond to positive veer angles above the hub and negative veer angles below the hub (veering), while negative values of $a_\varphi$ correspond to negative veer angles above the hub and positive veer angles below the hub (backing). Also, $a_\varphi = 0$ corresponds to zero veer, and values further from zero imply higher veer in the respective direction.

We assume that temperature modifies the air density and blade structural damping, with the first modeled using the ideal gas equation and International Standard Atmosphere. With regards to the variation of structural damping, studies on the material properties of composites similar to the ones used in wind turbine blades show that in general, the structural damping increases with the temperature (Spirnak and Vinson, 1990; Sefrani and Berthelot, 2006). However, the exact nature of the variation depends on the fiber material used in the blades and their orientation. In this work, a linear relationship between temperature and structural damping is assumed. The reference temperature for the normal damping values is considered as $15\,°C$, and a





decrease of 50% in the structural damping is assumed at $-10\,°C$. The variation of the damping of the first three modes with temperature is shown in Fig. 1.

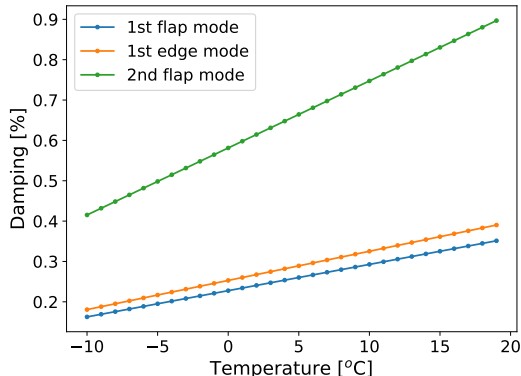

**Figure 1:** Variation of damping of the first three modes with temperature

In `HAWC2`, the structural damping is modelled via a stiffness-proportional damping model. As per this model, in the calculation of the damping matrix, factors $K_x, K_y, K_z$ are multiplied to moments of inertia of the element about the x,y and z axes,

$I_x, I_y$, and $I_z$, and inserted into the damping matrix. The details about the calculation of the damping matrix found in Hansen (2001). Typically the coefficients $K_x, K_y, K_z$ are tuned until the required structural damping of the first three blade modes is achieved. For a given value of the coefficients, and the structural model of the turbine blade, the structural damping can be quickly calculated using the program `HAWCStab2` (Hansen, 2011). To calculate the value of the coefficients for a given temperature, the new values of damping of the first three modes are first calculated, and the values of $K_x, K_y, K_z$ are tuned using

`HAWCStab2` until the new damping values for the first three modes are achieved. The tuning is framed as an optimization problem with a cost function $C = \sum_{i=1}^{3}(\zeta_i - \zeta_{i,req})^2$, where $\zeta_i$ represents the damping of the $i^{th}$ mode for a given value of the coefficients, and $\zeta_{i,req}$ represents the required damping for a given temperature. The optimization problem is solved using the framework `OpenMDAO` (Gray et al., 2019).

**SIV characterisation**

The edgewise bending moment at the root of blade 1 is chosen as the response to study SIV. A growth of edgewise bending moment with time implies the presence of SIV while a decay implies absence. The behaviour of the wind turbine for the different inflow conditions broadly falls into three categories: linearly stable, linearly unstable and non-linearly unstable (where the system enters a limit cycle after some time). The corresponding edgewise bending moments are shown in Fig. 2. It is generally observed that the responses where the system behaves as non-linearly unstable grow quicker than when the system

behaves as linearly unstable. Hence, these nonlinearly unstable scenarios are more negatively damped than the linearly unstable ones, implying more critical vibrations.



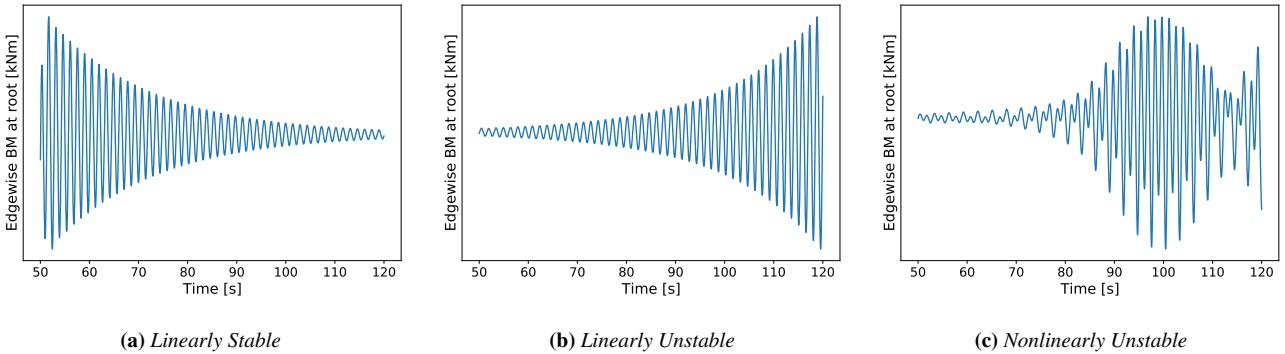

**Figure 2:** Typical responses of the wind turbine blade under different yaw angles.

The stability of the system with respect to SIV is characterised by the identifying the damping ratio of the first blade edge mode. The damping ratio is chosen over other measures based on the amplitude such as peak to peak amplitude or max. amplitude because the BEM simulations have a tendency to predict amplitudes that are not physical, but nonetheless qualitatively comparable to CFD based simulations. The first 50 s are discarded as transient. The response is limited in time to be in the linear range to make it suitable for a damping identification method. One of the most widely used methods to identify the damping ratio is the logarithmic decrement method. However, the classic logarithmic decrement method is sensitive to the choice of peaks, and different peaks in the same response can lead to identification of significantly different damping ratios. Hence, we have used a variant of the logarithmic decrement method ratio which avoids this problem by considering many response peaks, and hence is more robust.

The damping ratio is identified using this method as follows: The damped frequencies of the dominant blade mode(first edge mode in the case of SIV) can be identified as the frequency corresponding to the peak of the Power Spectral Density (PSD) of the response. This is justified in this analysis because we consider an isolated blade (stiff tower, hub and nacelle) with no turbulent inflow, and hence the peaks of the PSD of the response is expected to be around the damped modes of the blade. Let $\omega_d$ represent the damped frequency of the first blade edge mode. The response, band-pass filtered around $\omega_d$ can be written as

$$y(t) = Ae^{-\zeta\omega_n t}\sin(\omega_d t + \phi), \tag{2}$$

where $\zeta$ represents the mode damping ratio, and $\omega_n$ the mode natural frequency. However, the finite frequency resolutions, and in some cases, the spectral leakage during the calculation of the PSD poses challenges to the accurate identification of $\omega_d$ from the PSD peaks. So, to make the damping identification routine more robust, the response is bandpass-filtered around $\omega_n$. This is justified because $\omega_d$ and $\omega_n$ are related as $\omega_d = \omega_n\sqrt{1-\zeta^2}$, and for low values of $\zeta$, $\omega_d \approx \omega_n$. This approximation is valid in the context of SIV, which are not expected to have very high values of $\zeta$.

In Eq. (2), the term $Ae^{-\zeta\omega_n t}$ indicates the growth/decay of the peaks of the response , and so by indicating with $(t_i, p_i)$ the time and value of the $i^{th}$ response peak, it can be written that

$$\log(p_i) = -\zeta\omega_n t_i + c, \tag{3}$$





where $c$ is the line intercept. Thus, the mode damping ratio can be calculated from the slope of the straight line fitted to the logarithm of the bandpass filtered response peaks and the corresponding time instances. The straight line is fit with the least-squares method.

## 4 Surrogate Based Optimization Setup

The proposed framework to study SIV includes an experiment design, a simulation environment, a surrogate model and an
160 adaptive sample generation technique. A brief description of these components and the SBO process is given in this section.

### 4.1 Design Of Experiments

The initial samples for studying the parameter space are generated using a Latin Hypercube Sampling (LHS) method using the library `OpenTURNS` (Baudin et al., 2016). Such a sampling tends to distribute points all over the parameter space, avoiding potential clustering of points. To ensure that the initial samples contain points with a negative damping (presence of SIV), the
165 following method is used to generate the samples. Roughly, the following conditions are guessed favourable for SIV - high wind speeds, moderate yaw angles, and low temperatures. Accordingly, trapezoidal distributions are considered for wind speed, yaw, and temperature with high PDF values at high wind speeds, moderate yaw angles, and low temperatures respectively. The effect of $\alpha$ and $a_\varphi$ are not guessed beforehand, and hence a uniform distribution is considered for $\alpha$ and $a_\varphi$. A total of 100 initial samples are generated, of which 68 are generated using the LHS method, and 32 samples are generated by considering
the combinations of the extreme values of the inflow variables (vertices of the hypercube formed by the inflow space).

### 4.2 Surrogate Model

After generation of samples, `HAWC2` simulations are performed, and for each simulation the damping ratio is identified using the method described in section 3. Then, a surrogate model is trained on the dataset comprising of the initial samples and the corresponding damping ratios. The surrogate models establishes an input output relationship between the inflow variables and
175 the damping ratio. This way, the damping ratio can now be expressed as a continuous variable across the domain, which helps in identifying a minimum value via standard optimization techniques. After investigating the performance of many surrogate models, Gaussian Process Regression (GPR) models (Kriging) which have been widely used in SBO problems are considered here. GPR models are interpolation models, and hence take the exact function value at the known sample locations, which makes them an attractive choice to be used in such applications where each evaluation of the target function is the result of a
180 computationally expensive simulation. Also, in comparison to models which require a high training time like ANN, the GPR models take significantly lesser time to train, and this also makes them an attractive choice as the surrogates need to be trained multiple times in the proposed SBO framework. This part about training the surrogate multiple times is explained in detail in the forthcoming section 4.3.1 which describes the exploration part of the adaptive sampling strategy. In this work, a GPR model with a squared exponential kernel is used which is briefly described below. The GPR model is implemented using the
185 library `Scikit Learn`(Pedregosa et al., 2011). A brief description of the GPR model is as follows





Let $\mathbf{x}$ denote the independent variable in $N$ dimensions (in this case the five inflow space variables) and $y$ the dependent variable (in this case the damping ratio). Let $X = \{\mathbf{x}_1, \mathbf{x}_2, \mathbf{x}_3, ....\mathbf{x}_n\}$ denote the set of sample points, and let $y = \{y_1, y_2, ...y_n\}$ denote the value of the target function at the sample points. Let $S$ denote the surrogate model that is obtained using $X$ and $y$ as training data. The GPR model for $S$ uses a weighted sum of $n$ functions with each function centered about the corresponding sample point. GPR models assume that the underlying target function is the result of a Gaussian Process (GP). The use of GP to model the underlying target function has its roots in geo-statistics where it was originally used to estimate surface heights of a given terrain based on heights sampled at a few locations.

$$S(\mathbf{x}) = \sum_{i=1}^{n} \lambda_i \psi(d(\mathbf{x}, \mathbf{x}_i)), \tag{4}$$

where $\lambda_i$ are the weights (the model parameters), and $\psi$ is the squared exponential kernel that is centered around each point, and $d(\mathbf{x}, \mathbf{x}_i)$ denotes the distance between the points $\mathbf{x}$ and $\mathbf{x}_i$. The kernel is given by

$$\psi(d(\mathbf{x}_i, \mathbf{x}_j)) = \exp\left(-\frac{1}{2}\frac{d(\mathbf{x}_i, \mathbf{x}_j)^2}{l^2}\right) \tag{5}$$

where $l \geq 0$ is the length scale parameter. If $l$ is a scalar, the kernel is called an isotropic kernel, and if $l = \{l_1, l_2, l...l_N\}$ is a vector with a dimension N i.e. each independent variable has its own length scale, then the kernel is called an anisotrpic kernel. In that case, the distance between the points is calculated after scaling the points with the length scale in each dimension. In this work, an anisotropic kernel is used. The length scale parameters are obtained by Maximum Likelihood Estimation.

The input variables are normalized to the range [0,1] while the target variable is standardized by scaling to unit mean and zero standard deviation, and the GPR model is trained on the normalized data. The reason for normalizing the inflow variables to the range [0,1] is to facilitate easier estimation of length scales and avoid numerical ill-conditioning when calculating the covariance matrix which is dependent on the distance between sample points. The reason for standardizing the output variable is because GPR models assume a zero mean prior on the target data. And similarly, whenever a prediction is to be made using the trained model, the data points to be predicted are normalized to [0,1], and the resulting output is then inverse scaled to obtain the predicted damping ratio.

### 4.3 Adaptive sampling

Adaptive sampling is a strategy to generate subsequent samples in the SBO process. The SBO process has two phases: exploration and exploitation, and each phase has its own adaptive sampling strategy. Exploration is the process of improvement of understanding of SIV behaviour in the domain while Exploitation is the process of understanding critical areas in the domain. The exploration phase aims at improving the accuracy of the surrogate model globally, while the exploitation phase aims at reaching the global minimum damping in the domain. In both phases, the samples are chosen iteratively. The adaptive sampling strategy for both phases is explained below.



### 4.3.1 Exploration

In the exploration phase the aim is to improve the global performance of the surrogate model. One way to do this is to add more samples in the empty regions of the domain and in regions where target function or the surrogate $S_i$ is highly nonlinear. A way to identify empty regions in the input space is by Delaunay triangulation of the domain. Delaunay triangulation is a way to divide the domain into triangles (simplexes in higher dimensions) with the sample points as vertices such that no sample point lies inside the circumcircle of any triangle thus being formed. The Delaunay triangulation maximises the minimum angle of all the triangles and hence avoids formation of skewed triangles. The simplexes with high volumes indicate large empty regions in the domain. Once the large regions are identified, the problem of complexity is addressed by placing the sample at the 'most non-linear' point within the large simplex. This method of selecting samples using Delaunay triangulation and a measure of complexity is similar to the algorithm proposed by Garud et al. (2019b), and only slight modifications from the original algorithm have been adopted in this work. The samples in the exploration phase are selected in iterations or rounds, with $k$ samples per round. A description of the selection of $k$ new samples for one round is explained in the remainder of this subsection.

The Delaunay triangulation of the $N$ dimensional space formed by $X$ leads to $T$ Delaunay simplices $\{D_1, D_2, ...D_T\}$ with volumes $\{V_1, V_2, ...V_T\}$, circumcenters $\{C_1, C_2, ...C_T\}$, and circumradii $\{R_1, R_2, ...R_T\}$ respectively. The volume $V_t$, circumcenter, $C_t$ and circumradius $R_t$ of a simplex $D_t$ with known vertices can be calculated as

$$V_t = \sqrt{\left( \frac{(-1)^{N-1}}{2^N (N!)^2} |\mathcal{M}_t| \right)} \tag{6}$$

$$\mathcal{M}_t = \begin{bmatrix} 0 & 1 & 1 & \cdots & 1 \\ 1 & 0 & d_{12}^2 & \cdots & d_{1(N+1)}^2 \\ 1 & d_{21}^2 & 0 & \cdots & d_{2(N+1)}^2 \\ \vdots & \vdots & \vdots & \ddots & \vdots \\ 1 & d_{(N+1)1}^2 & d_{(N+1)2}^2 & \cdots & 0 \end{bmatrix} \tag{7}$$

where $d_{ij}$ denotes the distance between vertices $i$ & $j$, and $\mathcal{M}_t$ is called the Cayley-Menger matrix. $R_t$ in cartesian coordinates, and $C_t$ in Barycentric co-ordinates $\{\gamma_1, \gamma_2, \gamma_3, ..., \gamma_{N+1}\}$ can be obtained by solving the following equation

$$\mathcal{M}_t \begin{bmatrix} -2R_t^2 \\ \gamma_1 \\ \gamma_2 \\ \vdots \\ \gamma_{N+1} \end{bmatrix} = \begin{bmatrix} 1 \\ 0 \\ 0 \\ \vdots \\ 0 \end{bmatrix} \tag{8}$$

For the theorems related to calculating the volumes, circumcentres and cirucmradii, an interested reader is referred to Garud et al. (2019b). Equation (8) gives the circumcenter in barycentric co-ordinates, and it should be converted to cartesian coordi-





nates for use later. The cartesian co-ordinate of $C_t$ in the $i^{th}$ dimension can be calculated from its corresponding barycentric coordinate as

$$C_t^{(i)} = \sum_{j=1}^{N+1} \gamma_j x_j^{(i)} \tag{9}$$

where $x_j^{(i)}$ denotes the value in the $i^{th}$ dimension of the point $\mathbf{x}_j$.

From the Delaunay triangulation, the simplexes with the first $k$ high volumes are chosen, and the samples are to be placed inside these simplexes. Let $D^*$ denote one such simplex. Since $D^*$ is a simplex in $N$ dimensions, it has $N+1$ vertices $\{\mathbf{x}_{D1}, \mathbf{x}_{D2}, ...\mathbf{x}_{DN}\}$. The location of the sample point within $D^*$ is determined using a measure of complexity. To do this, first a linear approximation $L(\mathbf{x})$ is constructed using the $N+1$ points formed using the vertices of the simplex as independent variable values and the value of the target function at the vertices as the dependent variable value. Then the deviation function $\delta(\mathbf{x}) = |S_i(\mathbf{x}) - L(\mathbf{x})|$ gives a measure of the non-linearity. However, maximizing $\delta$ within $D^*$ may lead to the problem of the new sample point being too close to the vertices of the simplex, and hence may not help in effectively addressing the problem of empty regions. To address this, a distance function is defined as $\phi(\mathbf{x}) = \prod_{j=1}^{N+1} ||\mathbf{x}_{Dj} - \mathbf{x}||$ as the product of the new sample point from all the vertices of $D^*$. Then, the product $\delta(\mathbf{x}) \times \phi(\mathbf{x})$ is optimized to balance the complexity and distance to existing points. Finally, to make the optimization problem simpler, the optimization is done within the circumsphere of $D^*$ rather than $D^*$ itself, as it simplifies the definition of the bounds of the problem. While this does increase the search region beyond $D^*$ in some regions, this expansion can be argued as being better as we search a larger unexplored space. The final optimization problem that is solved is

$$\text{max. } \delta(\mathbf{x})\phi(\mathbf{x}) \tag{10}$$
$$\text{subject to } ||\mathbf{x} - C^*||^2 \leq R^{*2}$$

where $C^*$, and $R^*$ denote the circumcenter and circumradius of $D^*$ respectively. At the end of the optimization, if the optimization fails, the next sample is located at the centroid of $D^*$ whose coordinate in a given dimension is the average of the coordinate value of the vertices.

In each round, the metric used to measure the accuracy of the surrogate model is the $R^2$ value calculated using the Leave One Out approach, defined as

$$R^2 = 1 - \frac{\sum_{i=1}^{n} (y_i - \hat{y}_i)^2}{\sum_{i=1}^{n} (y_i - \bar{y}_i)^2}, \tag{11}$$

where $y_i$ represents the target function value at the $i^{th}$ sample, $\hat{y}_i$ refers to the corresponding prediction made by $S_i$ and $\bar{y}_i$ refers to the average value of the target function at all samples. This exploration metric provides a termination criteria for the exploration phase. The exploration phase is done until the $R^2$ value in the last $n$ consecutive iterations reaches an arbitrary threshold $\epsilon$. Here $n$ and $\epsilon$ are set to 3 and 0.8 respectively. The exploration process is summarised in Fig. 3.

Generate initial samples using LHS

Perform simulations

Identify Damping

Generate/Update database of all simulations and results

Fit surrogate models, and calculate $AE_i$ and $R^2$

$R^2 > \epsilon$ in the last $n$ rounds?

Yes → **Proceed to exploitation**

No

Tessellate the space using Delaunay triangulation method and identify the $k$ largest Delaunay simplexes

Within each simplex, is there a point $\mathbf{x}$ that satisfies eq. (10)?

No → Generate new samples at the centroid of the identified simplex

Yes → Generate new sample at $\mathbf{x}$

Generate $k$ such new samples

**Figure 3:** The Exploration process





### 4.3.2 Exploitation

In the exploitation phase, the aim is to find critical areas in the domain, i.e., areas of very low damping. As we try to study a larger dimension space with few points, and even after achieving satisfying accuracy of the surrogates in the exploration phase, the minima predicted by the surrogate model and the actual minima may not be the same. While the typical goal in SBO is to find the global minimum of the function, it is also advantageous to know the behavior of SIV in the parameter space requires knowledge of critical conditions all over the space than just the global minimum. To achieve this, we keep iteratively minimizing the surrogate model and identify $m$ minima less than a specified threshold each round. Then, at the identified $m$ minima, HAWC2 simulations are performed, and the process is repeated until the global minima converges. However, if the global minima does not converge after $N_{exploit}$ rounds, to obtain an estimate of the global minima with the information available, the following scheme is employed.

### 4.3.3 Convergence scheme

As we try to study a larger dimension space with few points, even after achieving satisfying accuracy of the surrogates in the exploration phase, the minima predicted by the surrogate model and the actual minima may not be not the same. This problem of the actual and predicted minima not being the same has been observed in previous works with SBO (Sessarego et al., 2016). In such cases, if an agreement is not achieved between the actual and the predicted minima after $N_{exploit}$ rounds, a new scheme is proposed to conclude convergence to a respectable minima. The basic idea with the scheme is that, in the neighbourhood of the predicted minima, the observed target function values should be low, which is a justified assumption for a continuous target function. For every predicted minima, the corresponding Delaunay simplex with the closest centroid is identified, and the value of the target function is evaluated at the vertices of the simplex. A threshold is then set on the average of the vertices values. The predicted minima that do not meet this criteria are regarded as possibly false, and are not considered as samples for the next round. The exploitation phase is repeated until the minima converges. In this work, $N_{exploit}$ is chosen as 20. The exploitation phase is summarised in Fig. 4.

### 4.4 Framework validation

The proposed framework is validated against the the six-hump camel back function and the Rosenbrock function. The camel back function is a 2 dimensional function given by

$$f(\mathbf{x}) = \left(4 - 2.1x_1^2 + \frac{x_1^4}{3}\right)x_1^2 + x_1x_2 + (-4 + 4x_2^2)x_2^2 \tag{12}$$

The function is usually defined in the domain $x_1 \in$ [-3,3], $x_2 \in$ [-2,2]. The camel back function has six local minima and two global minima. It is visualized in Fig. 5a. The global minima of $y$ = -1.0316 occurs at $\mathbf{x}^*$ = (0.0898, -0.7126) and (-0.0898, 0.7126).



**Figure 4:** The Exploitation process





The Rosenbrock function is a multidimensional function given by

$$f(\mathbf{x}) = \sum_{i=1}^{d-1}[100(x_{i+1} - x_i^2)^2 + (x_i - 1)^2] \tag{13}$$

The function is usually defined in the domain $\mathbf{x} \in$ [-5,10], and has a global minima of $y = 0$ at $\mathbf{x}^* = 1$. The minima lies in a narrow, parabolic, valley around $\mathbf{x}$=1. The Rosenbrock function in two dimensions is shown in Fig. 5b.

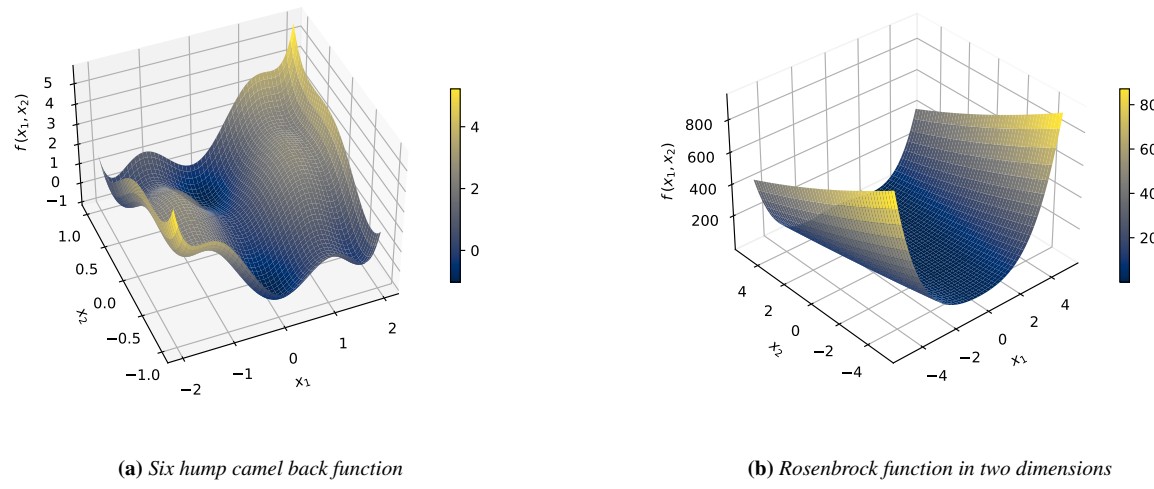

**(a)** *Six hump camel back function*        **(b)** *Rosenbrock function in two dimensions*

**Figure 5:** Test functions

The proposed framework is tested on the six-hump camel back function, and the 5-dimensional Rosenbrock function to see if the minima of these functions can be identified by the framework. To check for robustness against initial sampling, the process is done for three different sets of initial samples. The test functions are also evaluated on the popular SBO algorithm Efficient Global Optimization (EGO), with an adaptive sampling strategy based on the Expected Improvement (EI) function (Jones et al., 1998). The results of the identification of the minima with both the algorithms are shown in figures 7 and 6 respectively.

The EGO algorithm was implemented using the library `SMT` (Bouhlel et al., 2019).

     It can be seen that, for the 2-dimensional six-hump camel back function, the proposed framework requires more samples than the EGO algorithm to identify the minima, whereas for the 5-dimensional Rosenbrock function, for all the three different sets of initial sample sets considered, the EGO algorithm does not identify the true minima or achieve convergence, while the proposed framework converges consistently to the true minima of the function. The considered SIV problem involves 5

dimensions, and the performance of the framework in solving a similar 5 dimensional rosenbrock problem is an indication of the suitability of the framework for the considered SIV problem.




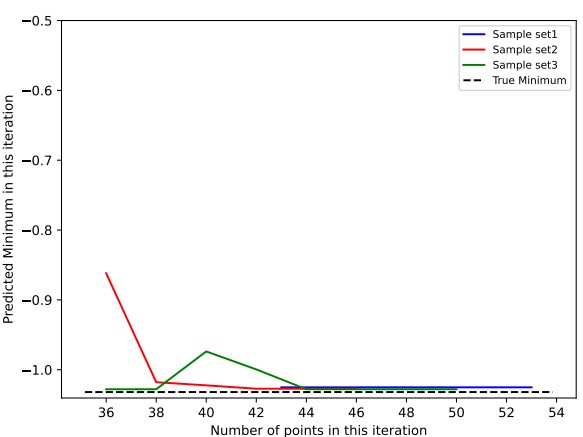

**(a)** *Exploitation phase of the proposed framework. Previous samples are a part of the exploration phase with 10 initial samples.*

**(b)** *EGO algorithm based on the EI adaptive sampling strategy and 10 initial samples.*

**Figure 6:** Identification of minima of the six-hump camel back function with three different sets of initial samples. To make the plot clearer, values greater than 3 are saturated to 3 and marked in square in the plots. The difference in the scales is to be noted.

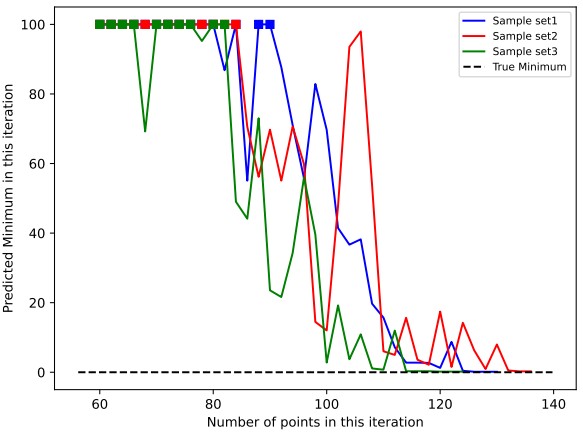
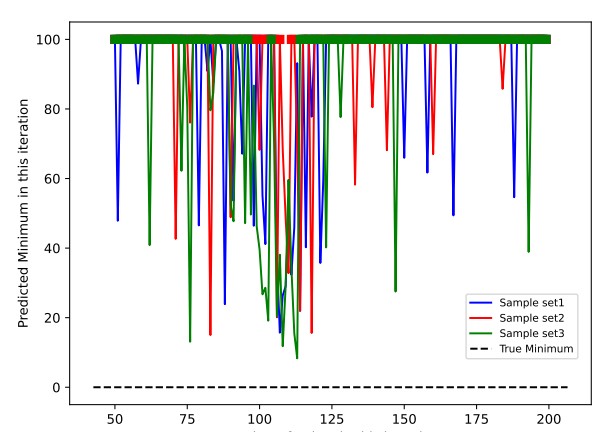

**(a)** *Exploitation phase of the proposed framework. Previous samples are a part the exploration phase with 50 initial samples.*

**(b)** *EGO algorithm based on the EI adaptive sampling strategy and 50 initial samples.*

**Figure 7:** Identification of minima of the Rosenbrock function with three different sets of initial samples. To make the plot clearer, values greater than 100 are saturated to 100 and marked in square in the plots.





## 4.5 Framework performance in analysing the SIV problem

The evolution of the surrogate accuracy and the evolution of minimum damping ratio is shown in Fig. 8. The most severe SIV as predicted by the proposed framework is around -6%. The corresponding inflow conditions are around Wind speed $= 60\,\mathrm{ms}^{-1}$, yaw angle = 18 deg, $a_\varphi$ = -1.2, $\alpha$ = 0.4, Temperature = $20\,°\mathrm{C}$ The convergence to the minima is shown in figure 9. Since there

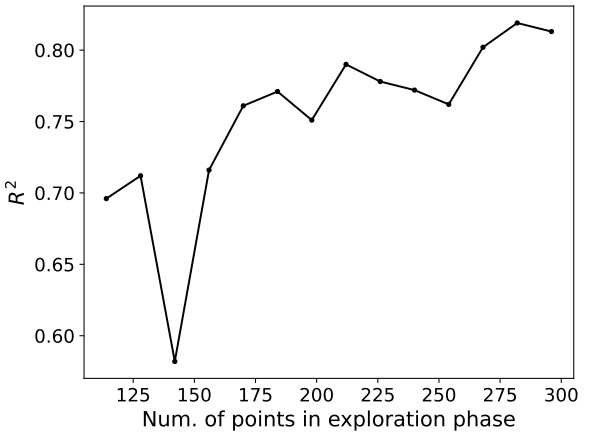

**(a)** *Evolution of Accuracy of the surrogate model*

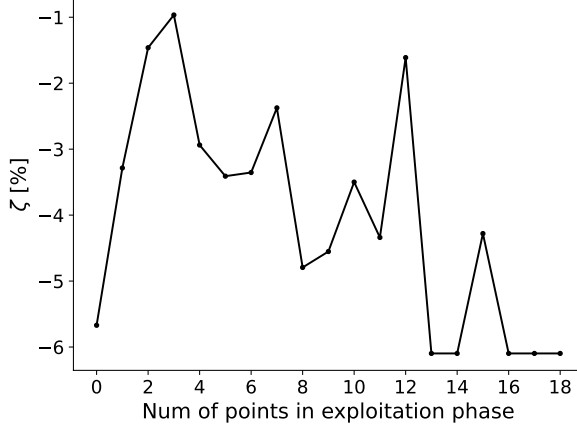

**(b)** *Evolution of Minimum Damping*

**Figure 8:** Performance of Framework.

are five dimensions involved, the convergence is shown individually for each variable.



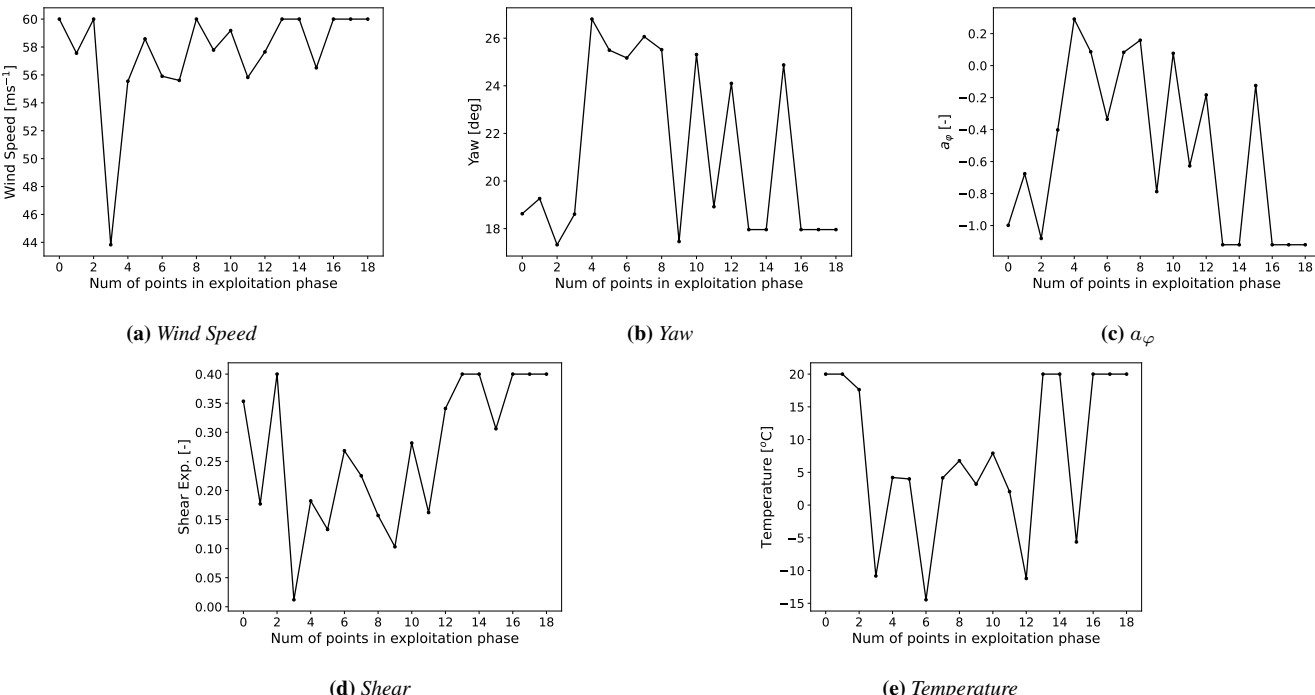

**(a)** *Wind Speed*  **(b)** *Yaw*  **(c)** $a_\varphi$

**(d)** *Shear*  **(e)** *Temperature*

**Figure 9:** Convergence to minima

## 5   Influence of variables on SIV

The influence of the input variables on SIV is studied using sensitivity analysis. A global sensitivity analysis can help to understand the overall influence of the input variables, while a regional sensitivity analysis can help to identify the combination of variables that lead to certain types of damping behaviour. This can then be used to identify inflow conditions that lead to severe SIV.

Sobol Indices, which measure the fraction of variance of each input variable (and also combinations) in relation to the total variance of the output variable can be used to measure global sensitivity (Saltelli et al., 2008). For an $n$-dimensional input space with $i, j \leq n$, the first order Sobol index $S_i$ indicates the effect of individual variable $i$, while the second order index $S_{ij}$ indicates the effect of interaction of variables $i$ and $j$, and so on for the higher indices, with a higher value indicating a higher influence. For an output variable $Y = f(X_i)$, the first order Sobol indices, $S_i$ are given by

$$S_i = \frac{\mathrm{Var}[E(Y \mid X_i)]}{\mathrm{Var}(Y)} \tag{14}$$

The second order indices are given by

$$S_{ij} = \frac{\mathrm{Var}[E(Y \mid X_i, X_j)]}{\mathrm{Var}(Y)} - S_i - S_j \tag{15}$$

and so on.





Sobol indices are always in the range [0,1]. The calculation of Sobol Indices involves a large number of evaluations of the output with respect to the inputs, and the surrogate model is useful here. Multiple function evaluations are done quickly using the surrogate model, and the indices are calculated using the python library `SALib` (Herman and Usher, 2017). The first order Sobol indices of the different variables are shown in table 2.

**Table 2:** First Order Sobol Indices

| Wind Speed | Yaw | Veer | Shear | Temperature |
|:---:|:---:|:---:|:---:|:---:|
| 0.02 | 0.425 | 0.026 | 1.68E-03 | 4.4E-04 |

It can be seen that the yaw angle is the most influential variable on SIV. Wind speed and veer have a mild influence, while shear and temperature have the least influence. The reason for the very low influence of temperature could be attributed to the fact that the effect of temperature is modelled through a change in structural damping, and for the turbine considered, the structural damping of the first edgewise mode itself is low (0.36%) as compared to the negative aerodynamic damping, which can be as high as $-6\%$. Therefore, changes in temperature so not affect the overall damping significantly.

The quantity $1 - \sum S_i$ indicates the effect of interaction between the variables. Here, $\sum S_i = 0.473$, which implies a still significant interaction between the variables. To get the influence of the second order interaction among the variables, the second order Sobol indices are calculated. They are shown in table 3.

**Table 3:** Second Order Sobol Indices

| | Wsp | Yaw | Veer | Shear | Temperature |
|:---|:---:|:---:|:---:|:---:|:---:|
| **Wsp** | | 0.146 | 0.006 | 0.02 | 0.001 |
| **Yaw** | 0.146 | | 0.138 | 0.009 | 0.002 |
| **Veer** | 0.006 | 0.138 | | 0 | 0 |
| **Shear** | 0.02 | 0.009 | 0.003 | | 0 |
| **Temperature** | 0.001 | 0.002 | 0 .001 | 0 | |

It can be seen that there are significant interactions between wind speed and yaw, and yaw and veer. The interaction with respect to temperature and shear are very low, as for the first order Sobol index. Thus, it can be said that for this setup, yaw

angle, wind speed and veer are the important variables in understanding the behaviour of SIV.

## 5.1 Occurrence

Using the surrogate model, the inflow conditions that lead to SIV are also studied using techniques of regional sensitivity analysis (Saltelli et al., 2008). The inflow space is divided into two regions based on the damping predicted using the surrogate





model. In region $B$, the damping ratio is negative ($\zeta \leq 0$), and in the other region $\bar{B}$, the damping ratio is positive. The
distribution of the variables in $B$ and $\bar{B}$ are then analysed, and compared using a standard test such as the 2-sided Kolmogorov-
Smirnov test (KS test) to analyse the similarity of the distributions. The test statistic for each test is $D_i = \sup \|F(X_i|B) - F(X_i|\bar{B})\|$ where $F$ represents the Cumulative Density Function (CDF). The CDFs of $X_i$ and the values of $D_i$ are shown in
figure 10.

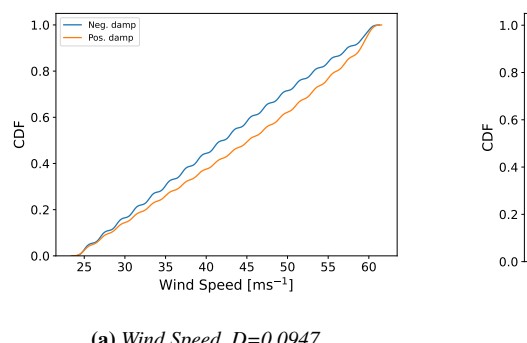 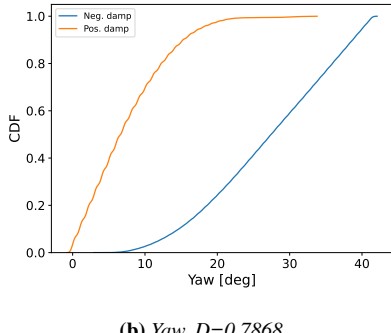 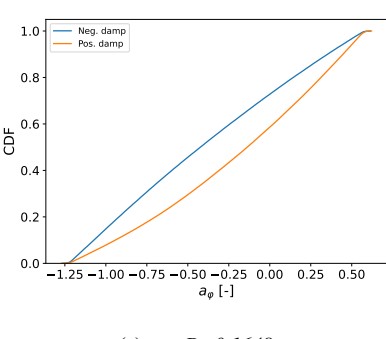

**(a)** *Wind Speed, D=0.0947*  **(b)** *Yaw, D=0.7868*  **(c)** $a_\varphi$, *D=0.1648*

**Figure 10:** CDFs and KS test statistics of the different variables.

For each $X_i$, the steeper regions of the CDF of $B$ represent the values that lead to the output belonging to $B$. Also, a larger
value of $D_i$ indicates that the CDFs are significantly different, and hence if $X_i$ is influential in deciding if the output lies in the
region $B$. From Fig. 10, it can be seen that yaw angles with a large $D$ are the most influential in the occurrence of a negative
damping. The yaw angle range around $[10, 40]$ deg is the most conducive for a negative damping to happen. Wind speed and
$a_\varphi$ have a mild influence in deciding the occurrence of SIV as the respective $D$ values are low, and the distributions are still
separated. As expected from the Sobol index values, the $D$ values of shear and temperature are close to zero (not shown in
figure 10), indicating that no specific values of these variables are favourable for SIV to occur.

## 5.2 Severity

The conditions that lead to severe SIV are analysed in a similar way, but now we define $B$ as the region where $\zeta$ is severely
negative. In this work, a scenario is severely negative, if $\zeta \leq -3\%$. The analysis of how the inflow space variables are distributed
in $B$ can help in understanding the conditions that lead to severe SIV. Figure 11 shows the distribution of the inflow space
variables in $B$.

It can be seen that the most severe SIV are predicted to occur at high wind speeds, and yaw angle in the range $[10, 30]$ deg.
It can also be seen that the distributions of the other variables span all over their defined ranges. So, it is predicted that severe
negative damping can occur at any possible value in their range, although positive shear and negative veer are more probable
to lead to severe SIV.

An analysis of the joint occurrence of the variables can provide additional information about the interaction between the vari-
ables in $B$. Since wind speed and yaw angle seem to be the most restrictive in terms of defining the region $B$, the combination





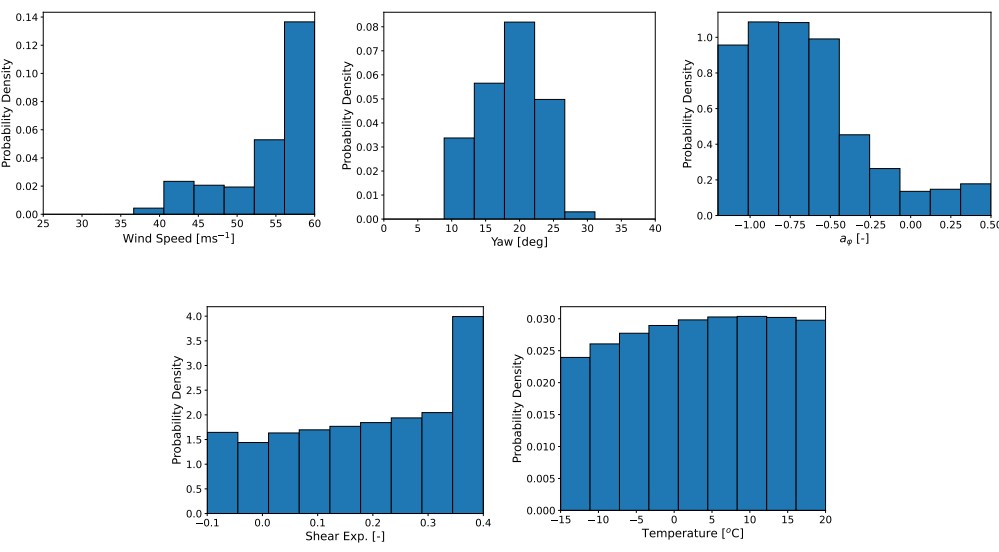

**Figure 11:** Distribution of variables in regions that lead to severe negative damping.

with one of these variables is expected to provide the most information. The joint occurrence of the variables with yaw angle is shown in figure 12. It can be seen that the joint densities with wind speed and $a_\varphi$ are concentrated in specific regions and also indicate some trends. For example, it can be seen that the yaw angles at which severe SIV occurs increases with increasing veer

in the positive direction. Also, the probability of observing severe SIV is higher at negative veer than positive veer, indicating that situations with negative veer are more conducive than situations with positive veer for severe SIV to happen. Similarly, severe SIV are more probable at higher wind speeds. The joint densities with shear and temperature are relatively spread all over, indicating that the variables do not have a significant impact on the severity. In addition to identifying regions leading to severe damping ratio, the surrogate model can be used to get estimates of the damping ratio for severe conditions, as shown in

Santhanam et al. (2022)

It is to be noted that the severity of SIV as described in this work is measured using the damping ratio at low amplitudes of the edgewise bending moment. However, the simulations are performed using a BEM-based aeroelastic solver which tends to over-predict the instabilities, and predict limit cycles which are unlikely to happen in reality. Hence a more accurate picture of the instabilities are expected using a high cost, high fidelity CFD based aeroelastic solver. But nonetheless, the methods

presented here can help as a guiding tool to narrow the inflow conditions to be simulated using the CFD solvers, thereby optimizing the computational cost required to explore the inflow space.





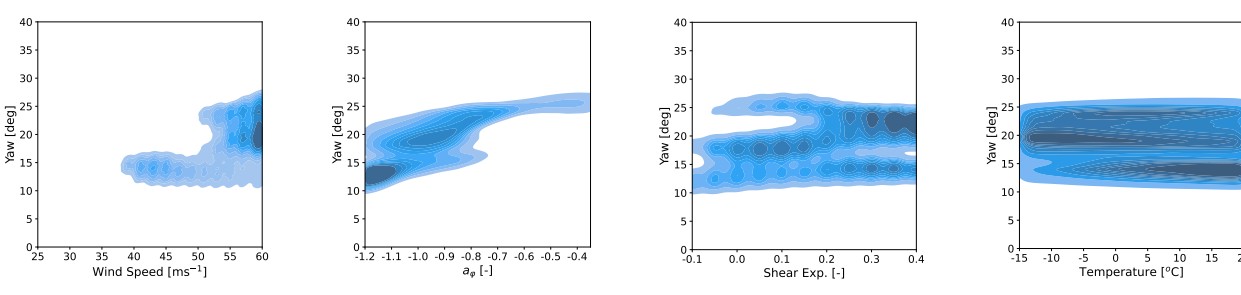

**Figure 12:** Joint densities of the variables in the severely negatively damped region.

## 6 Conclusion and future works

A surrogate-based optimization framework has been developed to study instabilities in wind turbines and demonstrated to study the effect of multiple inflow variables on SIV. In the application of the framework, the damping ratio of the first blade edgewise mode was considered as the stability measure, and the framework was used to identify the most influential variables, and combination of inflow variables that lead to critical SIV on the IEA 10 MW turbine. Though only environmental variables have been included in the application presented in this paper, the framework can be extended to include design variables such as parameters related to the structural and aerodynamic characteristics of the blade, which can be useful in getting a quick estimate of critical conditions and effect of turbine design changes.

Since the proposed framework is not limited to any particular type of surrogate model, use of surrogate specific features have not been explored to get an estimate of uncertainty of the predictions. Development of methods to estimate uncertainties without too much exploitation of surrogate specific features would be an improvement to the proposed framework. As a future work, the proposed framework is to be used to study other types of aeroelastic instabilities such as Vortex-Induced Vibrations (VIV).

*Acknowledgements.* This work has been supported by the PRESTIGE project (J.no. 9090-00025B), granted by Innovation Fund Denmark.

*Author contributions.* CS conducted the simulations, invented and developed the optimization framework and did the main writing of the paper. RR and TK discussed all aspects of this work and edited the manuscript.

*Competing interests.* The authors declare that there are no competing interests.



*Code availability.* Code is available upon request



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
