# Peer review of "A Surrogate Based Optimization Framework to analyse Stall Induced Vibrations"

_Wind Energy Science, 2022_

## Referee Comment (RC1)

Manuscript ID: WES-2022-79

Title: A Surrogate Based Optimization Framework to Analyze Stall Induced Vibrations

1.  Comments

    ●   The authors proposed a surrogate-based optimization framework. However, there is nothing new in the framework. All the strategies used in the frame are already available. So, it seems to the reviewer that there is no academic novelty in the manuscript at least regarding surrogate aspect.

    ●   The manuscript title implies that the authors applied the framework for optimization. However, I can't find much about the optimization results obtained applying the proposed framework.

    ●   It is not well explained why the surrogate approach is necessary for the given problem. At least, the authors should explain the computer simulation time and compare the total computation time with and without using the surrogate model for the given problem.

2.  Conclusion and Suggestion

    The paper is not recommended for publication in its present form.

---

## Referee Comment (RC3)

**Review**:

**Manuscript Title: A Surrogate Based Optimization Framework to analyse Stall Induced Vibrations**

**Paper: wes-2022-79**

In this study, a surrogate-based optimization framework is implemented to explore the behavior of the wind turbine blades' stall-induced vibrations (SIV). SIV is vital for large and flexible wind turbine blades and needs to be accounted for during the design studies. However, due to the high computational cost associated with the aeroelastic simulations, the presented surrogate-based approach for optimization accounting SIV is significant to the wind energy community.

The authors have implemented Gaussian Process Regression (Kriging) model as surrogates and implemented the Delaunay Triangulation method to select the samples and improve the accuracy of the surrogates. As a case study, the IEA 10MW turbine was selected with five variables to define the inflow conditions- wind speed, yaw angle, vertical wind shear, wind veer, and atmospheric temperature. Furthermore, the sensitivity analysis was also carried out to identify the influential variables. Based on the results, the yaw angle was found to be the most significant variable, whereas the temperature had the least influence on SIV.

The manuscript is well-written and includes the appropriate references. However, the manuscript can be improved by addressing the following comments. Therefore, the reviewer recommends for **Major Revision** of the manuscript in its current form.

The comments that might help improve the manuscript are given below:

**Major Comments**

1. The computational cost associated with each simulation or response function (damping ratio) evaluation should be included.
2. The threshold used for the termination of the exploration phase is provided as $\epsilon = 0.8$ in Line 266. Generally, the value of $R^2$ used for surrogate modeling is 0.95-0.99. The author needs to provide reasoning for selecting this value of 0.8.
3. The sentences in Section 4.3.3, line 284-286, are unclear.

   > *"For every predicted minima, the corresponding Delaunay simplex with the closest centroid is identified, and 285 the value of the target function is evaluated at the vertices of the simplex. A threshold is then set on the average of the vertices values. The predicted minima that do not meet this criteria are regarded as possibly false, and are not considered as samples for the next round."*

   a. What does false mean here? Does it mean inaccurate? The reasoning for not utilizing the already evaluated responses at these predicted minima even though it does not satisfy the threshold needs to be appropriately explained.
4. This study presents a sampling approach based on exploration and exploitation by utilizing the Delaunay triangle, which is one of the main contributions, as mentioned in lines 61-63. The performance of this approach has been compared with expected improvement-based

EGO. It would be better for the readers to see the actual comparison of the main application problem related to the SIV of wind turbine blades presented in this study.

5. Multiple runs of the presented approach are provided for analytical problems; however, it is not provided for the main application, which is the optimization regarding the SIV. The algorithm's robustness to initial samples and runs should also be demonstrated for the main application problem.

6. The authors should also shed some light regarding the non-monotonic convergence of the damping ratio in Figure 8-b. For example, what optimizer (algorithm) was used, and does the non-monotonic convergence trend depend on the optimizer?

7. In Section 5, the influence of variables on SIV is studied using Sobol Indices-based global sensitivity analysis. While the first-order and second-order Sobol indices are provided, it's recommended to also include the total order Sobol Indices that includes information regarding the individual and mixed-order interactions/ contributions of the input variables.

**Minor Comments:**

1. The sentence in Line 268 is repeated exactly in Line 278. Need to paraphrase.
   *"As we try to study a larger dimension space with few points, even after achieving satisfying accuracy of the surrogates in the exploration phase, the minima predicted by the surrogate model and the actual minima may not be the same."*

---

## Author Comment (AC1)

**Response to referee 1's comments**

Chandramouli Santhanam, Riccardo Riva and Torben Knudsen

November 4, 2022

The authors thank the reviewers for the constructive comments and suggested improvements. A revised version of the paper has been prepared considering the reviewers' comments. A list of replies to the reviewers' comments is reported below.

**RC1**

**RC1 a)**

**The authors proposed a surrogate-based optimization framework. However, there is nothing new in the framework. All the strategies used in the frame are already available. So, it seems to the reviewer that there is no academic novelty in the manuscript at least regarding the surrogate aspect.**

A) Novelties of our paper:

The authors acknowledge that there is a limited novelty from the point of developing an Surrogate-Based Optimization (SBO) strategy itself such as a new surrogate model, or optimization methods as that is not the scope of this work. But it is the view of the authors that designing an SBO framework using different existing strategies and applying it to the Stall-Induced Vibration (SIV) problem is a new work (to the best of the authors' knowledge).

This point are now mentioned in the last paragraph of the Introduction which has been edited to read, "To the authors' best knowledge, the problem of effectively exploring the behaviour of SIV using an SBO framework has not been attempted before and in this paper, we propose an SBO type framework to study SIV effectively. The proposed SBO framework is independent of the surrogate type, and in addition to identifying the most critical vibrations, has a wider scope to additionally identify regions in the domain where the target function can take critical values (below a certain threshold) with good confidence."

B) Differences with respect to existing strategies

Delaunay Triangulation has been used previously in SBO applications for adaptive sampling,

for ex. in [1],[2],[3],[4]. The authors acknowledge and mention in section 4.3.1 that most of the ideas used in this paper are available in the existing literature, especially [1] & [2].

To elucidate the key differences between these (and other similar works) and this paper, the following line has been removed from section 4.3.1

" [1]"

And the following lines that explain the differences with respect to existing strategies are now added after explaining the framework, at the end of section 4.3.3.

"The method of selecting samples using Delaunay triangulation and a measure of complexity is similar to the algorithm proposed by [1], and the method of iteratively minimizing the surrogate model in the exploitation phase is similar to the method used in [2]. The key differences between these (and other similar works) and this work are:

1. In the exploration phase, we use a $R^2$ metric after every iteration as a measure of the quality of the surrogate and to decide when to stop the exploration phase, while this metric has not been proposed in the works that have used Delaunay triangulation for adaptive sampling. This metric serves two purposes - a) Track the progress of the exploration phase and b) To decide the termination of the exploration phase.

2. In the exploitation phase, if the minima cannot be found after a specified number of iterations, we propose a scheme based on the properties of the Delaunay simplexes to eliminate false predicted minima and conclude the process to a respectable minima using the information available (section 4.3.3). The previous studies that have used the method of iterative minimization of the surrogate model to identify the minimum have not used such a method to eliminate false minima in case convergence is not reached."

**RC1 b)**

**The manuscript title implies that the authors applied the framework for optimization. However, I can't find much about the optimization results obtained applying the proposed framework**

The authors thank the reviewer for providing the perspective of the optimization part from a neutral reader's perspective. The authors would like to convey that sections 4.3.2, 4.3.3, and 4.4 are related to the optimization of the surrogate model, and figure 8b shows the results of the minimum damping after each iteration. To emphasize the optimization used in this work, section 4.3.2 has been rewritten to mention the multi-start minimize method that has been used in the optimization process

**RC1 c)**

**It is not well explained why the surrogate approach is necessary for the given problem. At least, the authors should explain the computer simulation time and**

**compare the total computation time with and without using the surrogate model for the given problem**

The authors thank the reviewer for pointing out that the paper lacks a mention about the usefulness of surrogate models. To address this lack, the following lines are added towards the end of section 5.

The effectiveness of the surrogate-based framework can be seen in the number of simulations necessary to study SIV in this domain with and without the surrogate based approach. Without the surrogate based framework, to generate the information presented in this section, simulations need to be conducted at atleast 10 levels each for wind speed, yaw angle and temperature, and 5 levels for shear and veer each. Assuming a full factorial design to study SIV in the domain, this leads to a total number of $10^3 \cdot 5^2 = 25000$ necessary simulations. With the surrogate based framework, however, the behaviour of SIV in the domain is analysed with just a few hundred simulations. The trade-off is the uncertainty introduced by the surrogate model but as it can be seen, the computational benefit is huge.

**References**

[1] S. S. Garud, N. Mariappan, and I. A. Karimi, "Surrogate-based black-box optimisation via domain exploration and smart placement," *Computers and Chemical Engineering*, 2019.

[2] E. Davis and M. Ierapetritou, "A centroid-based sampling strategy for kriging global modeling and optimization," *AIChE journal*, vol. 56, no. 1, pp. 220–240, 2010.

[3] Y. Wu, L. Ozdamar, and A. Kumar, "Triopt: a triangulation-based partitioning algorithm for global optimization," *Journal of computational and applied mathematics*, vol. 177, no. 1, pp. 35–53, 2005.

[4] P. Beyhaghi, D. Cavaglieri, and T. Bewley, "Delaunay-based derivative-free optimization via global surrogates, part i: linear constraints," *Journal of Global Optimization*, vol. 66, no. 3, pp. 331–382, 2016.

---

## Author Comment (AC2)

**Response to referee 2's comments**

Chandramouli Santhanam, Riccardo Riva and Torben Knudsen

November 4, 2022

The authors thank the reviewers for the constructive comments and suggested improvements. A revised version of the paper has been prepared considering the reviewers' comments. A list of replies to the reviewers' comments is reported below.

**RC2**

**RC2 a)**

**The major problem is that HAWC2 is applied to analyze a parked rotor. Note that the verification article the authors cited applies to operating wind turbines with rotating blades. To the reviewer, the validity of the results is subjected to questions. Since surrogate models are applied, why not using CFD or a lifting line model?**

The authors acknowledge that Blade Element Momentum (BEM) theory is not applicable for a rotor in standstill. In this work, the aerodynamic model used in HAWC2 is the Near Wake Model (NWM) which has been implemented in HAWC2 and compared against analytical solutions and measurements for standstill conditions [1].

The verification article for HAWC2 has been changed to [1].

The information about the NWM model has been included in the paper. The second paragraph of the first section has been rewritten as

"SIV have been studied using the Blade Element Momentum theory (BEM) based solvers in the limited yaw angle range around moderate stall regions. The Advanced Aerodynamic Tools for Large Rotors (AVATAR) project details a comprehensive SIV study of BEM-based aeroelastic solvers against a higher fidelity CFD based aeroelastic solver [2]. The results show that BEM-based solvers tend to over-predict standstill instabilities due to the utilization of static airfoil data. But since the basic assumption in BEM theory that the rotor can be modelled as a disc does not hold in standstill conditions, other models such as the Near Wake Model (NWM) have been proposed for aerodynamic modelling in standstill conditions [1]. The Near Wake Model has been validated against analytical solutions for an elliptical wing and measurements against the NREL Phase VI rotor in standstill conditions. While the

NWM model deviates from the measurements in certain conditions, it is still a better choice than the Blade Element Momentum (BEM) model."

The reason for not using CFD or lifting line methods in spite of using surrogate models is that the time and effort required are still high for an initial domain exploration. This point is included at the end of the first paragraph of section 3 as

"A HAWC2 is preferred over CFD simulations because of the computational time involved. While a typical HAWC2 simulation takes around 20 minutes, CFD simulations typically take much larger computational time. Additionally, the complexity of setting up the simulations is higher than HAWC2 simulations. For an initial domain exploration, it is advantageous to use solvers like HAWC2, which is still costly for a 5-dimensional problem. The initial exploration results can help decide the focus of higher fidelity CFD simulations and lifting line methods that can be used for a detailed study of the instabilities."

**RC2 b)**

**Table 1 shows a number of considered variables. Please clarify why they are selected here and whether there are other more important parameters**

The variables in Table 1 are chosen because the wind speed, yaw angle, shear and veer affect the aerodynamic damping, and temperature affects the structural damping. There are more parameters related to environmental conditions, and blade geometry affecting SIV. We have chosen five parameters taking into account the computational complexity. Of course, the framework can still be used to study the problem effectively, but we have limited the number of dimensions to 5 for a demonstration. The following lines have been added at the end of section 2.
"While there are other parameters that affect SIV, the number of variables is limited to five due to computational considerations".

**RC2 c)**

**Some of the statements are not justified. For example, "The reference temperature for the normal damping values is considered as 15 °C, and a decrease of 50% in the structural damping is assumed at -10 °C" Are there any grounds that can be used to support this statement?**

The authors thank the reviewer for pointing out the necessity to justify the consideration about the assumed effect of temperature better. The authors would like to say that to the best of their knowledge, the exact nature and the amount of variation of structural damping of composite materials with temperature depends on the material type and orientation of the fibres. The amount of variation considered in this study have been reported in glass fibre composites, for example in figure 4 in [3].

This information is now included in the paper. The following lines have been added at the end of the third paragraph of section 3.

"While the exact variation of structural damping with temperature depends on the fibre material used in the blades and their orientation, glass fibre composites have been reported to show the amount of variation considered in this study [3]."

**RC2 d)**

**The language is acceptable, but a check should be done to avoid a mixed use of American/British spelling in the manuscript**

The authors thank the reviewer for this comment and have corrected the document to avoid mixed use of American/British spelling.

**References**

[1] G. R. Pirrung, H. A. Madsen, and S. Schreck, "Trailed vorticity modeling for aeroelastic wind turbine simulations in standstill," *Wind Energy Science*, vol. 2, no. 2, pp. 521–532, 2017.

[2] J. Heinz, N. N. Sørensen, V. Riziotis, M. Schwarz, S. Gomez-Iradi, and M. Stettner, "Aerodynamics of large rotors wp4 deliverable 4.5," Tech. Rep. D4.5, ECN Wind Energy, Petten, The Netherlands, August 2016.

[3] H. Zhen, W. Xiang, X. Xiao-lin, L. Zhi-Peng, and W. Li-Ying, "Study on glass fiber/epoxy gradient damping composites," *Asian Journal of Chemistry*, vol. 25, no. 7, p. 3831, 2013.

---

## Author Comment (AC3)

**Response to referee 3's comments**

Chandramouli Santhanam, Riccardo Riva and Torben Knudsen

November 4, 2022

The authors thank the reviewers for the constructive comments and suggested improvements. A revised version of the paper has been prepared considering the reviewers' comments. A list of replies to the reviewers' comments is reported below.

**RC3**

**RC3 a)**

**The computational cost associated with each simulation or response function (damping ratio) evaluation should be included.**

The authors thank the reviewer for this comment that would improve the case for using surrogate models. The computational cost associated with each HAWC2 simulation is included in the first paragraph of section 3 by adding the following lines

"While a typical HAWC2 simulation takes around 20 minutes, CFD simulations typically take much larger computational time. Additionally, the complexity of setting up the simulations is higher than HAWC2 simulations. For an initial domain exploration, it is advantageous to use solvers like HAWC2, which is still costly for a 5-dimensional problem. The initial exploration results can help decide the focus of higher fidelity CFD simulations and lifting line methods that can be used for a detailed study of the instabilities. "

**RC3 b)**

**The threshold used for the termination of the exploration phase is provided as $\epsilon = 0.8$ in Line 266. Generally, the value of $R^2$ used for surrogate modelling is 0.95-0.99. The author needs to provide reasoning for selecting this value of 0.8**

The authors acknowledge that the threshold of 0.8 is rather low, but the considered problem is in 5 dimensions, and surrogate model is trained on a few points (of the order of $10^2$). Hence the surrogate model is expected to have accuracy in this range, and having an accuracy of the order of 0.95-0.99 would require a lot more simulations and consequently, more computational effort.

The following line has been added at the end of section 4.3.1

"The threshold on $R^2$ is set with the consideration that the considered problem is in five dimensions and the number of points is of the order of $10^2$. A higher threshold would require a higher number of simulations and hence incur a higher computational cost. "

**RC3 c)**

The sentences in Section 4.3.3, line 284-286, are unclear. "For every predicted minima, the corresponding Delaunay simplex with the closest centroid is identified, and 285 the value of the target function is evaluated at the vertices of the simplex. A threshold is then set on the average of the vertices values. The predicted minima that do not meet this criteria are regarded as possibly false, and are not considered as samples for the next round."
a. What does false mean here? Does it mean inaccurate? The reasoning for not utilizing the already evaluated responses at these predicted minima even though it does not satisfy the threshold needs to be appropriately explained

The authors thank the reviewer for pointing out the unclear parts of the paper.

The first line of section 4.3.3 has been modified to read

"Because the surrogate model is only an approximation to the true target function, the minimum value predicted by the surrogate model and the value of the target function at the predicted minima may not be the same, and thus the surrogate model predicts many 'false' minima. The predicted minima are false in the sense that the value of the target function at these points are not as low as predicted by the surrogate model"

The sentences of the first paragraph towards the end have been modified to read

"For every predicted minima, the corresponding Delaunay simplex with the closest centroid is identified, and the value of the target function is evaluated at the vertices of the simplex, and the average is calculated ($f_{D,\text{avg}}$). A threshold is then set on $f_{D,\text{avg}}$. The predicted minima whose $f_{D,\text{avg}}$ is higher than the threshold are regarded as possibly false, as they are surrounded by points with a high average value of the target function. Such predicted minima are not considered as samples for the next round."

**RC3 d)**

This study presents a sampling approach based on exploration and exploitation by utilizing the Delaunay triangle, which is one of the main contributions, as mentioned in lines 61-63. The performance of this approach has been compared with expected improvement-based EGO. It would be better for the readers to see the actual comparison of the main application problem related to the SIV of wind turbine blades presented in this study.

The EGO algorithm has been tried on the main application problem, and the results are presented in figure 10. An accompanying paragraph with the figure has been added at the

end of section 4.5

"The EGO algorithm has been run only for 20 iterations from the initial 100 samples but it is to be noted that the EGO algorithm identifies one new sample per iteration, and hence takes a much longer time compared to the framework proposed in this work which identifies multiple samples per iteration. hence, while the proposed framework and the EGO algorithm have been run for nearly the same amount of time, the EGO algorithm uses much fewer samples. Also, the focus of the EGO algorithm is to identify the global minimum, while the focus of the framework presented in this work is to also obtain a surrogate model that identifies multiple critical regions in the domain. Thus the feature that the proposed framework uses many samples is of great advantage since the surrogate model is trained better."

**RC3 e)**

**Multiple runs of the presented approach are provided for analytical problems; however, it is not provided for the main application, which is the optimization regarding the SIV. The algorithm's robustness to initial samples and runs should also be demonstrated for the main application problem.**

The authors thank the reviewer for this suggested enhancement which would show the robustness of the proposed framework better. The framework is tested on a different initial sample set, and the results are shown in the appendix.

**RC3 f)**

**The authors should also shed some light regarding the non-monotonic convergence of the damping ratio in Figure 8-b. For example, what optimizer (algorithm) was used, and does the non-monotonic convergence trend depend on the optimizer?**

The following lines have been added in section 4.3.2 to include information about the optimizer that has been used.

"Many choices for the minimization algorithm exist, and in this work the minimization was performed using the Sequential Least SQuares Programming (SLSQP) algorithm [1]. The minimization using SLSQP algorithm is implemented using the Python library `SciPy` [2]."

The reason for the non-monotonic trend of the convergence is that, as stated in section 4.3.3 of the paper, the minimum predicted by the surrogate model and the actual minima obtained from simulations is not the same.

This point has been included in the first paragraph of section 4.5 which now reads

"It can be seen that during the convergence, the minimum does not keep decreasing sequentially with iterations. The reason for this non-monotonic trend is that, as mentioned in section 4.3.3, the minimum damping ratio predicted by the surrogate model and the actual damping ratio from HAWC2 simulations do not always agree."

**RC3 g)**

**In Section 5, the influence of variables on SIV is studied using Sobol Indices-based global sensitivity analysis. While the first-order and second-order Sobol indices are provided, it's recommended to also include the total order Sobol Indices that includes information regarding the individual and mixed-order interactions/ contributions of the input variables.**

The authors thank the reviewer for this recommendation that would help in an enhanced inference of the Sobol indices. The total Sobol indices have been included in table 4 in section 5, and the following sentence has been added at the end of section 5.

"This can also be seen from the values of the total Sobol indices shown in table 4 which include the combination of first, second and all the other higher order indices"

**References**

[1] D. Kraft, "A software package for sequential quadratic programming," *Forschungsbericht-Deutsche Forschungs- und Versuchsanstalt fur Luft- und Raumfahrt*, 1988.

[2] P. Virtanen, R. Gommers, T. E. Oliphant, M. Haberland, T. Reddy, D. Cournapeau, E. Burovski, P. Peterson, W. Weckesser, J. Bright, S. J. van der Walt, M. Brett, J. Wilson, K. J. Millman, N. Mayorov, A. R. J. Nelson, E. Jones, R. Kern, E. Larson, C. J. Carey, İ. Polat, Y. Feng, E. W. Moore, J. VanderPlas, D. Laxalde, J. Perktold, R. Cimrman, I. Henriksen, E. A. Quintero, C. R. Harris, A. M. Archibald, A. H. Ribeiro, F. Pedregosa, P. van Mulbregt, and SciPy 1.0 Contributors, "SciPy 1.0: Fundamental Algorithms for Scientific Computing in Python," *Nature Methods*, vol. 17, pp. 261–272, 2020.